# The Antiviral Potential of AdipoRon, an Adiponectin Receptor Agonist, Reveals the Ability of Zika Virus to Deregulate Adiponectin Receptor Expression

**DOI:** 10.3390/v16010024

**Published:** 2023-12-22

**Authors:** Daed El Safadi, Grégorie Lebeau, Jonathan Turpin, Christian Lefebvre d’Hellencourt, Nicolas Diotel, Wildriss Viranaicken, Pascale Krejbich-Trotot

**Affiliations:** 1Unité Mixte Processus Infectieux en Milieu Insulaire Tropical, Plateforme Technologique CYROI, Université de la Réunion, INSERM U1187, CNRS UMR 9192, IRD UMR 249, 94791 Sainte Clotilde, La Réunion, France; daed.el-safadi@univ-reunion.fr (D.E.S.); gregorie.lebeau@univ-reunion.fr (G.L.); jonathan.turpin@univ-reunion.fr (J.T.); 2UMR 1188 Diabète Athérothombose Réunion Océan Indien (DéTROI), Campus Santé Université de la Réunion, Université de La Réunion, INSERM, 77 Avenue du Docteur Jean-Marie Dambreville, 97410 Saint-Pierre, La Réunion, France; christian.lefebvre-d-hellencourt@univ-reunion.fr (C.L.d.); nicolas.diotel@univ-reunion.fr (N.D.)

**Keywords:** Zika virus, antiviral, AdipoRon, adiponectin agonist, drug repositioning, metabolism

## Abstract

Zika virus (ZIKV) is a pathogenic member of the flavivirus family, with several unique characteristics. Unlike any other arbovirus, ZIKV can be transmitted sexually and maternally, and thus produce congenital syndromes (CZS) due to its neurotropism. This challenges the search for safe active molecules that can protect pregnant women and their fetuses. In this context, and in the absence of any existing treatment, it seemed worthwhile to test whether the known cytoprotective properties of adiponectin and its pharmacological analog, AdipoRon, could influence the outcome of ZIKV infection. We showed that both AdipoRon and adiponectin could significantly reduce the in vitro infection of A549 epithelial cells, a well-known cell model for flavivirus infection studies. This effect was particularly observed when a pre-treatment was carried out. Conversely, ZIKV revealed an ability to downregulate adiponectin receptor expression and thereby limit adiponectin signaling.

## 1. Introduction

Zika virus (ZIKV) is a mosquito-borne virus which, on the eve of the global crisis caused by SARS-CoV-2, had already alerted the international scientific community. The awareness was due to an unanticipated emergence, an ability to spread rapidly worldwide and, above all, a capacity to cause severe and previously unknown neurological diseases in newborns [1,2]. ZIKV at the time of its re-emergence, after successive epidemics, first in Micronesia in 2007, then French Polynesia in 2013, followed by Brazil in 2015, was not totally unknown [3]. This flavivirus is a member of the Flaviviridae family, which was first isolated in a rhesus monkey from the Zika rainforest in Uganda in 1947, indicating the existence of an African and an Asian lineage [4,5]. Both are mainly transmitted by mosquitoes of the genus Aedes. While the historical African strain of ZIKV seemed to cause only discrete and sporadic cases, the Asian strain has suddenly emerged as a threat. This urged the World Health Organization (WHO) to declare this virus a major health issue in 2015. Among flaviviruses, several are of medical concern, including Dengue Virus (DENV), Japanese Encephalitis Virus (JEV) and West Nile Virus (WNV). As with the latter, ZIKV infection often is asymptomatic in approximately 80% of cases and induces only mild symptoms in most patients. Unfortunately, contemporary epidemics revealed that ZIKV could also induce severe neurological diseases like Guillain–Barré syndrome in adults and congenital ZIKA syndromes (CZS) in infants [6,7]. Conversely to other flaviviruses, ZIKV has also shown the ability to bypass vector-based transmission through sexual and maternal–fetal transmissions, providing an additional mode of dissemination for this virus [8]. These specific features accentuate the risks taken by pregnant women or those planning to give birth in a ZIKA epidemic and endemic zones. To date, there is no treatment for ZIKV infection and no way to prevent the neurological damage that could occur in infected fetuses during pregnancy. Microcephaly, the most striking manifestation of fetal infection leading to CZS, is a phenomenon characterized by a decreased brain size. It has been shown that ZIKV can deregulate genes involved in the cell cycle, neurogenesis and apoptosis in the cortical progenitor cells and related neural progenitor cells. For key cells in brain development, the disruption of cell cycle progression, reduced proliferation, premature differentiation or increased mortality are highly detrimental [9,10]. In this context, the search for a molecule with a neuroprotective and antiviral role adapted to this particular situation of pregnancy becomes a priority.

Adiponectin is a cytokine produced by adipose tissue that is attracting growing interest due to its multiple physiological properties [11]. It was first identified for its anti-inflammatory effects and its ability to increase insulin sensitivity. Adiponectin is thus implied in a variety of metabolic diseases such as diabetes, atherosclerosis and dyslipidemia. Adiponectin signaling depends on two main types of adiponectin receptors, AdipoR1 and AdipoR2, which are ubiquitously expressed in a wide range of human tissues [12,13]. Adiponectin can be found in monomeric and multimeric forms. Its multimers include trimers, hexamers and higher-molecular-weight oligomers (12–18 mers, >300 kDa) [14]. The different forms, including high-molecular-weight complexes, circulate in the plasma and the trimeric and hexameric forms are found in human cerebrospinal fluid (CSF). Adiponectin in the CSF would benefit the glial and neuronal cells in oxidative-stress-, inflammation- and apoptosis-dependent pathological processes [15,16,17]. Adiponectin signaling in the central nervous system (CNS) has been shown to be involved in the regulation of brain homeostasis, neurogenesis, synaptic plasticity and memory improvement [18,19,20]. It is therefore considered neuroprotective and of therapeutic interest for various CNS disorders such as depression [21,22], stroke [23,24] and neurodevelopmental diseases [25]. AdipoR expression in the neurons of the hypothalamus, brainstem, cortex and the brain blood vessels supports the role of adiponectin signaling in the brain [26]. Moreover, adiponectin was shown to promote neurogenesis and cell protection, mainly via AdipoR1 signaling [27].

Due to its pleiotropic protective properties, adiponectin has been proposed as a pharmacological approach for the treatment of several pathologies. However, the high molecular weight of the most active form, its short half-life and stability have compromised its potential for clinical use [28], and directed research toward easier-to-use agonist molecules. AdipoRon is a synthetic small molecule that was the first orally active adiponectin receptor agonist to be discovered [29,30]. Several studies have shown its beneficial effects on brain functions and neurological disorders [31,32]. AdipoRon offers a number of pharmacological advantages linked to its ability to cross the blood–brain barrier. It has been detected in the brain following both oral administration and after systemic injection [32,33]. A low dose of AdipoRon promoted hippocampal cell proliferation and increased the serum adiponectin levels in mice [34]. Indeed, modulation of adiponectin signaling, whether by injection of adiponectin or AdipoRon, enhanced neuronal cell survival in various in vivo and in vitro models, including cerebral ischemia, cerebral hemorrhage and oxygen and glucose deprivation experiments [24,35,36,37,38]. The results include effective neuroprotection, a reduced infarct size and improved neurological outcomes. In addition, AdipoRon has been shown to compensate for the depletion of neural stem cells in the hippocampus by activating the AdipoR1/AMP-activated protein kinase (AMPK) pathway [21].

Based on the deleterious impact of ZIKV on neural progenitor cell proliferation and differentiation, we hypothesized that targeting adiponectin signaling could be an interesting therapeutic avenue for limiting ZIKV-induced damages in fetuses. Therefore, AdipoRon, which might provide a beneficial neuroprotective effect during ZIKV infection, emerged as a relevant, easy-to-use candidate, with already tried-and-tested therapeutic delivery options and efficacy [39,40,41,42]. To conduct an in vitro study of the effect of AdipoRon treatment on ZIKV infection, we used the A549 epithelial cells. This cell model had the advantage of already being well characterized for its permissiveness to ZIKV [43], and was then widely validated for the in vitro testing of pharmaceuticals and antiviral compounds [44,45,46]. We first verified that these cells expressed the adiponectin receptors. We then studied whether commercial purified adiponectin and its pharmacological analog, AdipoRon, could modify the outcome of in vitro infection. We also showed that AdipoRon treatment, at a non-cytotoxic dose, was able to promote the expression of adiponectin receptors in A549 cells. Surprisingly, this ability was reduced by ZIKV in infected cells. However, pre-treatment and, to a lesser extent, co-treatment with AdipoRon revealed promising antiviral effects in A549 epithelial cells infected with ZIKV.

## 2. Materials and Methods

### 2.1. Virus, Cell Culture and Reagents

Human epithelial A549^Dual^ cells (ATCC, CCL-185, Manassas, VA, USA) were cultured in DMEM (Dulbecco’s Modified Eagle Medium from Gibco/Invitrogen, Carlsbad, CA, USA) that contained 10% heat-inactivated Fetal Bovine Serum (FBS, good purchased from Invitrogen), 1 mmoL·L^−1^ sodium pyruvate, 2 mmoL·L^−1^ L-Glutamine, 0.1 mg·mL^−1^ of streptomycin, 100 U·mL^−1^ of penicillin and 0.5 µg·mL^−1^ of fungizone (PAN-Biotech, Aidenbach, Germany). The cells were cultured at 37 °C and under a 5% CO_2_ atmosphere. For ZIKV, a clinical isolate of the Asian lineage, isolated during the French Polynesian outbreak, PF-25013-18 (ZIKV^PF13^), was used and has been previously described [47]. The AdipoRon (SML0998) and adiponectin (1065-AP-050) were purchased from Sigma-Aldrich (Saint-Louis, MO, USA) and RnD Systems (Minneapolis, MN, USA), respectively. The AdipoRon was dissolved in absolute ethanol and adiponectin in sterile water as recommended by the manufacturers.

### 2.2. Lactate Dehydrogenase Assay (LDH)

The A549^Dual^ cells were plated in a 96-well plate with a density of seeding corresponding to 10,000 cells per well. The cells were then treated with increasing doses of AdipoRon (ranging from 0 µg·mL^−1^ to 100 µg·mL^−1^/233 µM). Then, 24 h later, the supernatants were collected. DMEM–dimethyl sulfoxide (DMSO) 10% was used as a positive death inductor.

LDH assay was achieved using the CytoTox 96^®^ Non-Radioactive Cytotoxicity Assay from Promega following the manufacturer’s recommendations. Briefly, 50 µL of LDH was added to 50 µL of LDH substrate and incubated at 37 °C for 15 min. The absorbance was then read at 490 nm using a FLUOstar^®^ Omega (BMG LABTECH, Offenburg, Germany).
$$cytotoxicity\;(\%)=100\times \frac{Experimental\;LDH\;release\;(OD490)}{Maximum\;LDH\;release\;(OD490)}$$

### 2.3. Neutral Red Assay

The A549^Dual^ cells were plated in a 96-well plate with a density of seeding corresponding to 10,000 cells per well. The cells were then treated with increasing doses of AdipoRon (ranging from 0 µg·mL^−1^ to 100 µg·mL^−1^). DMEM–DMSO 10% was used as a positive death inductor. Then, 24 h later, the cells underwent a neutral red assay.

The cell viability was assessed with a neutral red uptake assay, as described by Repetto et al. [48]. Briefly, 100 µL of medium containing 40 µg·mL^−1^ of neutral red was added to each well after the removal of the cell culture supernatant. The cells were then incubated for 2 h at 37 °C. Following incubation, the medium was removed, and the cells were washed with PBS. Finally, 150 µL of neutral red destain (50% ethanol, 49% H_2_O, 1% glacial acetic acid) was added per well. The absorbance was read at 540 nm using a FLUOstar^®^ Omega microplate reader (BMG LABTECH, Offenburg, Germany).
$$Cell\;viability\;(\%)=100\times \frac{Experimental\;condition\;(OD540)}{Negative\;control\;(OD540)}$$

### 2.4. Cell Infection and Treatments with AdipoRon or Adiponectin

The potential effect of AdipoRon treatment on the outcome of ZIKV infection was assessed on the A549 cells. Briefly, after seeding, cells were infected for 24 and 48 h with ZIKV^PF13^ at a given multiplicity of infection (MOI 1, 2 or 5). A condition of pre-treatment was performed, with AdipoRon added to the cells 16 h before infection (AdipoRon pre-ZIKV). A co-treatment was realized with AdipoRon added simultaneously with the virus inoculum (AdipoRon + ZIKV). One treatment was realized by adding AdipoRon 24 h post-infection, for 24 h (AdipoRon 24 h post-ZIKV). Another treatment was performed only during the last 8 h of infection (AdipoRon 40 h post-ZIKV). Infection was monitored using flow cytometry following ZIKV-E-positive cells, using the 4G2 antibody. The cell culture supernatants were harvested and stored at −80 °C, while the cells were lysed using RLT Lysis Buffer (QIAGEN, Hilden, Germany) for subsequent RNA extraction.

### 2.5. Flow Cytometry and Antibody

For the flow cytometry assay, the cells were gently harvested using trypsinization, fixed with 3.7% FA in PBS for 10 min and the 4G2-positive cells were analyzed using a CytoFLEX flow cytometer (Beckman Coulter, Villepinte, France). For viral detection, a primary mouse 4G2 antibody (anti-flavivirus E protein) from RD-Biotech (Besançon, France) was used at 1:1000 in PBS-BSA 1%. Donkey anti-mouse Alexa Fluor 488 IgG was used as the secondary antibody at 1:1000 in PBS-BSA 1%.

### 2.6. RNA Extraction and qRT-PCR

The total RNA was extracted from the cell lysates using the RNeasy Plus Mini Kit (cat. 74136, QIAGEN, Hilden, Germany). The total cDNA was obtained using reverse transcription and random primers from Invitrogen (ref. 58875, Thermo Fisher, Waltham, MA, USA) and an M-MLV reverse transcriptase enzyme (ref. M1708, Promega, Madison, WI, USA) at 42 °C for 60 min. Next, the cDNAs were subjected to a quantitative polymerase chain reaction, using a CFX96 Connect™ Real-Time Detection System (Bio-Rad, Hercules, CA, USA). For amplification, ABsolute™ 2X qPCR MasterMix (ref. AB1163A, Thermo Fisher, Waltham, MA, USA) and specific primers were used to assess gene transcript expression for the following targets: *GAPDH, RNAPII* and *RPLP0* for housekeeping genes and *ADIPOR1*, *ADIPOR2*, *ZIKV-E* (ZIKV envelope) and *CPT1* for targets of interest. The primers used are listed in Table 1. A threshold cycle (Ct) was calculated for each single sample amplification reaction in the exponential phase of amplification, using Bio-Rad CFX Manager 3.1 (Bio-Rad, Hercule, CA, USA). The qPCR datasets were analyzed using the ΔΔCt method (Livak and Schmittgen, 2001) and the results were normalized to the values obtained for the housekeeping genes.

### 2.7. Statistical Analyses

Statistical analyses were performed using one-way ANOVA with Dunnett’s correction using GraphPAD Prism software version 9. All values are expressed as mean ± SD of three independent experiments. Values of *p* < 0.05 were considered statistically significant. The degrees of significance are indicated as follows in the figure legends: ns = not significant, * *p* < 0.05; ** *p* < 0.001; *** *p* < 0.0002; **** *p* < 0.0001.

## 3. Results

A549 cells are an epithelial cell model that has been one of the most widely used to understand flaviviruses infection processes. Our team and others have characterized the in vitro infection of A549 cells by ZIKV [43,48,49]. This model has been used extensively to decipher cellular responses to infection, and in particular to assess the cytopathic effects induced by ZIKV [50,51,52]. It is particularly suitable and widely used for testing the in vitro effects of antivirals [46,53]. The literature’s data on adiponectin receptor expression by A549 cells are patchy [54]. Thus, we first checked whether these cells indeed expressed adiponectin receptor genes, to validate our cellular model for studying the effect of AdipoRon on ZIKV in vitro infection.

### 3.1. A549 Cells Express Both Adiponectin Receptor Genes, ADIPOR1 and ADIPOR2

Although AdipoR1 and AdipoR2, the main receptors for adiponectin, are known to be ubiquitously expressed [54,55,56], we first ensured that it was the case in our A549 cell model of ZIKV infection. We showed that both *ADIPOR1* and *ADIPOR2* are expressed, with *ADIPOR1* expression six-fold higher than that of *ADIPOR2* (Figure 1).

### 3.2. Assessment of the Non-Toxic Concentration of AdipoRon in A549 Cells

Given the concentration-dependent dual properties of AdipoRon on cell survival, and in order to test its pharmacological use, we treated cells with increasing concentrations of AdipoRon for 24 h. We estimated the cell cytotoxicity assays using cell leakage measurement (LDH activity in cell supernatants) and neutral red uptake, which measures the integrity of the endocytosis and lysosome pathways. We covered a spectrum of concentrations, from 3 to 100 µg·mL^−1^ (Figure 2). The 50% cytotoxic concentration (CC50) was calculated from the dose–response curve of viability obtained using the neutral red assay (Figure 2B). CC50 is estimated at 55 µg·mL^−1^ corresponding to 128.5 µM. We then considered 25 µg·mL^−1^, which corresponds to 58 µM of AdipoRon being the highest non-cytotoxic concentration we could assess on the A549 cells. When tested for 48 h of treatment, this concentration did not result in excess mortality (Figure 3E), indicating that it can be used over 24 and 48 h of infection.

### 3.3. AdipoRon Displays Antiviral and Cytoprotective Effects

We therefore infected A549 cells with ZIKV^PF13^, (i) pre-treating them with AdipoRon (25 µg·mL^−1^) 16 h before infection, (ii) co-treating them all along the 24 and 48 h of infection, (iii) post-treating them for 24 h after 24 h of ZIKV infection and (iv) post-treating them for 8 h, 40 h after the onset of infection (Figure 3A).

At both 24 and 48 h post-infection, a treatment with AdipoRon for 16 h before infection with ZIKV^PF13^ resulted in a significant two-thirds reduction in the percentage of infected cells, immuno-detected for the envelope protein (ZIKV-E) (Figure 3B,C). In addition, 48 h of co-treatment with AdipoRon led to an almost two-fold reduction in the percentage of infected cells (Figure 3C). A similar reduction was observed when the AdipoRon treatment took place 24 h after infection. However, a treatment applied 40 h after ZIKV infection for 8 h had no significant impact on the number of infected cells (Figure 3C).

To confirm the inhibitory effect of AdipoRon on ZIKV replication, we quantified the viral RNA in the infected and treated cell extracts using qRT-PCR. We observed a significant decrease in ZIKV RNA levels when the cells were pre-treated, co-treated or treated with AdipoRon 24 h after incubation with ZIKV (Figure 3D), which is consistent with the reduction in the percentage of infected cells assessed using cytometry. AdipoRon cell treatment prior to infection led to the lowest ZIKV infection as observed in Figure 3C,D. The cytotoxicity was analyzed by measuring the amount of LDH released from the cells, indicating that only a pre-treatment with AdipoRon for 16 h prior to infection led to significant cell protection (Figure 3E).

### 3.4. Adiponectin, like Its Pharmacological Analog AdipoRon, Has Anti-ZIKV Properties

To test whether the observed antiviral effect of AdipoRon is a property shared with adiponectin, we tested the effect of commercially purified adiponectin upon A549 cell infection with ZIKV^PF13^ at MOI 2. Based on the known concentrations of adiponectin circulating in body fluids [26] and after verification of its innocuity on cultivated A549 cells (Figure 4A), we incubated cells with adiponectin at 5 µg·mL^−1^. As with AdipoRon, adiponectin showed an ability to reduce cell infection when a 16 h treatment was applied to the cells prior to incubation with ZIKV. In contrast, co-treatment or the addition of adiponectin 24 h post-infection had no substantial effect on the percentage of ZIKV-E-positive cells detected 48 h post-infection (Figure 4B).

### 3.5. ZIKV Downregulates ADIPOR1 and CPT1 Gene Expression

To investigate the involvement of adiponectin signaling in limiting the infectious process, we analyzed the expression levels of genes in the AdipoR1/AMPK pathway under different conditions of ZIKV infection and AdipoRon treatment. In this axis, AMPK leads to the activation of p38 mitogen-activated protein kinase (MAPK), which in turn activates the peroxisome proliferator-activated receptor alpha (PPARα). PPARα is a transcription factor that upregulates the expression of genes associated with peroxisomal and mitochondrial β-oxidation, including the carnitine palmitoyltransferase 1 (*CPT1*) gene. Indeed, it has been previously demonstrated that adiponectin, by sequentially activating AMPK and p38 MAPK, increases the PPARα activity in the muscle cells [57] (Figure 5A). The increased transport of fatty acids into the mitochondria will result in their increased aerobic catabolism in response to adiponectin. The expression levels were analyzed using qRT-PCR for *ADIPOR1* and *CPT1*, the signaling target known to be upregulated upon adiponectin transduction and stimulation of the AdipoR1/AMPK/PPARα pathway.

Unexpectedly, while treatment with AdipoRon led to the upregulation of both *ADIPOR1* and *CPT1* expression, we observed that the transcriptional expression of both genes was halved under a 48 h ZIKV infection condition (Figure 5B). Upregulation of *ADIPOR1* was maintained under the AdipoRon pre-treatment conditions, which may account for the protective effect obtained on cells subsequently infected with ZIKV. When added simultaneously with ZIKV or after 24 h of infection, AdipoRon appeared to slightly attenuate the inhibitory effect of ZIKV on the expression of both genes. To ascertain ZIKV’s ability to inhibit the adiponectin receptor expression, we infected A549 cells with ZIKV^PF13^ at different MOIs and monitored the effect on both *ADIPOR1* and *ADIPOR2* expression after 24 h of infection. We confirmed that the *ADIPOR1* expression was downregulated by half as early as 24 h post-infection, independently of the MOI used (Figure 5C). Furthermore, we found that the downregulation also affected *ADIPOR2* to the same extent.

## 4. Discussion

Following worldwide Zika epidemics in the 2006–2016 period, and given the explosion of reported cases of adverse outcomes of pregnancies, congenital syndromes and birth defects (including about 2000 cases of ZIKV-linked microcephaly during the Brazilian outbreak [58], ZIKV must be considered a dangerous teratogenic pathogen [59]. Under these conditions, the search for antiviral drugs that can be administered to pregnant women living in ZIKV-endemic regions has become a challenge. This search for safe, bio-active molecules capable of protecting the developing CNS of the fetus is indeed a necessity, especially since vertical transmission from mother to child is possible and deleterious at various stages of pregnancy [60], including the possibility of an early contamination via the father’s semen [61]. This research is also needed to anticipate the possibility of the reemergence of the ZIKV Asian strains that still circulate silently in several world spots [62,63,64,65], but also in response to the threat of emergence of contemporary African strains whose teratogenic capacities are under the spotlight [66]. Unfortunately, and even if the literature on the discovery of antivirals selected from high-throughput screening of compound libraries; from observation of the traditional pharmacopoeia usages, notably based on plant and marine biodiversity [46,67] or from de novo design, is plethoric, it has to be admitted that, to date, few therapeutic strategies have resulted in conclusive clinical phases [68,69]. Among the huge efforts to find suitable prophylactics and treatments, drug repositioning offers several undeniable advantages. The proven safety and knowledge of the best drug delivery options are major strengths for the particular situation of ZIKV exposure and maternal–fetal health [70]. One specificity in the case of ZIKV is that the profile of a good drug candidate will be based on its ability to cross the placental and blood–brain barriers to reach and act on the fetus while being completely safe for its development. There are several possible strategies. A first approach is to test molecules with proven antiviral activity, already validated and already used for other viral infections. A potentially pan-viral curative treatment regimen combining drugs targeting several proteins (the viral replicase and proteases) of the hepatitis C virus (HCV), a Flaviviridae member related to ZIKV, has been developed with success [71,72]. Another approach would be to turn to molecules with cytoprotective effects used in CNS pathologies not necessarily of infectious origin. This is why we considered adiponectin, and more specifically its pharmacological analog AdipoRon, as a potential candidate of interest. These molecules, already known to exert neuroprotective activity, fit the essential criteria set out above [15,32]. Additionally, AMPK dysregulation and metabolic dysfunction are found in neurodegenerative processes, and the activation of the AdipoR1/AMPK pathway by AdipoRon has demonstrated therapeutic benefits in neuroinflammation following intracerebral hemorrhage [36,73]. Hypothesizing that the cytoprotective action of AdipoRon could favorably influence the outcome of ZIKV infection, we found in our in vitro infection model of A549 cells expressing adiponectin receptors (mainly *ADIPOR1* and to a lesser extent *ADIPOR2*, Figure 1) that AdipoRon was indeed able to limit infection. Pre-treatment of the cells at a concentration of 25 µg·mL^−1^, validated as non-cytotoxic (Figure 2), reduced by 2/3 the number of replicating ZIKV cells, and the associated cytopathy (Figure 3). While the treatment of cells with AdipoRon leads to the overexpression of the gene encoding AdipoR1 and that of CPT1, a transcriptional target of PPARα activated during adiponectin signaling, we revealed that ZIKV was able to interfere with this signaling pathway (Figure 5). Building upon prior studies [57], the observed downregulation of *CPT1* indicates that a potential regulation of AMPK activity, or of one or more of its downstream targets in the adiponectin/AdipoR pathway, may have occurred during infection. Experiments to measure the level of expression or activation of these factors should help to determine the effect of ZIKV on this signaling pathway.

We demonstrated ZIKV’s ability to halve the *ADIPOR1* and *ADIPOR2* expression as early as 24 h post-infection. This suggests that the adiponectin signaling pathway and its amplification loop are detrimental to ZIKV. To our knowledge, we have shown for the first time that the adiponectin signaling pathway has antiviral potential, making it a target that ZIKV subverts to limit its effect. The mechanisms underlying the deregulation of *ADIPOR1* and *CPT1* transcriptional expression by ZIKV deserves to be further investigated. The literature’s data on the regulation of AdipoR expression mainly mention the controls exerted by metabolic status and insulin [12,74]. The role of metabolic reprogramming induced during ZIKV infection on AdipoR1/R2 expression then needs to be clarified. The direct activity of viral proteins cannot be ruled out. Therefore, it would be interesting to investigate whether viral factors would be more likely to lead to transcriptional repression [75].

In our study, we highlighted that the ability of AdipoRon to limit infection when administered in pre-treatment is a feature shared by adiponectin. However, considering the lesser effect observed with adiponectin compared to its pharmacological analog, we speculate that AdipoRon may mimic the most active form of adiponectin in its anti-ZIKV capacity. This form, yet to be identified, would be weakly represented in the population of various multimers possibly present in the commercial suspension of adiponectin we used. The adiponectin doses may need to be increased to provide efficacy similar to that of its analog. Nevertheless, we do not exclude that AdipoRon may have activities independent of its ability to mimic adiponectin. However, the literature does not mention any related effects of this molecule, currently characterized exclusively for its “adiponectin receptor agonist” effects.

Finally, we still need to understand the cellular and molecular mechanisms by which adiponectin signaling exerts an inhibitory effect on ZIKV multiplication. Several hypotheses can be formulated to investigate the links between the identified cytoprotective capacities of adiponectin signaling and cellular activities that might impair viral multiplication. Among the possibilities to be explored, metabolic reprogramming is one of the most exciting. On the one hand, AdipoR1/AMPK signaling has been shown to be responsible for a cellular shift toward increased oxidative metabolism. The data available in the literature repeatedly mention that AdipoR signaling favors mitochondrial biogenesis and activity, oxidative phosphorylation (OXPHOS) and ATP synthesis through peroxisome proliferator-activated receptor gamma coactivator 1-alpha (PGC-1α) activation and its downstream targets [37,76,77,78]. On the other hand, it was demonstrated that viruses benefit from glycolytic cellular metabolism [79]. Conversely, aerobic metabolism and high OXPHOS are associated with a better interferon response and antiviral properties [80,81]. It therefore appears that viruses have strategies to favor the glycolytic use of glucose instead of OXPHOS, to increase viral production and limit antiviral response. In this regard, ZIKV and a few other flaviviruses are known examples of viruses that promote metabolic reprogramming to their benefit [82,83,84,85,86]. Indeed, the AMPK system is a sensor of cellular energy status that responds primarily to changes in the AMP:ATP ratio and activated OXPHOS. This diaphonic relationship between the reprogramming of the cellular metabolism and infection could therefore be disrupted by agonists of the AdipoR1/AMPK/PPARα axis. Potentiation of an antiviral response is not the only lever on which AdipoR1 agonists could act. The potentiation of cytoprotective mechanisms could also explain the beneficial effects of AdipoR1 agonists. Of note, some works describe how adiponectin-mediated heme oxygenase-1 (HO-1) induction could contribute to the cytoprotective effects attributed to adiponectin signaling [87]. HO-1 has been qualified as antiviral against several viruses [88], including ZIKV, but with underlying mechanisms that are not known. Interestingly, in a previous work, we demonstrated that HO-1 was a cellular antiviral player that ZIKV was able to subvert [89]. Setting these data in perspective suggests that adiponectin signaling and HO-1 antiviral activity may be linked. The cyto-protection associated with signaling by adiponectin and the AdipoR1/AMPK pathway is also attributed to a capacity to inhibit inflammation and oxidative stress via inhibition of the activity of the transcription factor nuclear factor kappa B (NF-κB) [11]. AMPK inactivates the P65 subunit of NF-κB by stimulating the expression of the gene encoding the “silent information regulator sirtuin 1” (SIRT1), which deacetylates P65. This SIRT1 expression is mediated by the transcription factor PGC-1α, activated by the AdipoR1–AMPK axis [90,91]. Another mechanism of regulation by AMPK is the inhibition of p65 phosphorylation by targeting IKKb, leading to the absence of nuclear translocation of p65 [92]. It should be noted that several studies describe antiviral drugs that target cellular factors, notably AMPK. PF-06409577, an AMPK activator that leads to modifications in the host cell lipid metabolism, was shown to impede the replication of various flaviviruses such as ZIKV [93]. AdipoRon induces AMPK activation, via the AdipoR1 signaling pathway (Figure 6). As with PF-06409577, the specific reduction in levels of various lipids important for flavivirus multiplication, subsequent to this signaling pathway and an activated AMPK, could be at the origin of viral restriction [94].

We highlighted in our study that the effect of AdipoRon or adiponectin is limited once infection has set in. This limitation may be due to the mechanism of action of these agonists and/or to the virus’ ability to downregulate the expression of the adiponectin receptors, as we stated (Figure 6). With the hypothesis that AdipoR1/R2 agonists enhance OXPHOS, it is worth remembering that infection, as discussed above, promotes aerobic glycolysis. AdipoR agonists and the virus therefore act in opposition for metabolic reprogramming. Indeed, it has been shown that M1 macrophages that display low OXPHOS have a lower expression of *ADIPOR1/R2* than those of the M2 phenotype exhibiting high OXPHOS [95,96]. Then, one explanation for the decrease in *ADIPOR1/R2* expression following ZIKV infection could be the virus-induced metabolic reprogramming toward aerobic glycolysis. These metabolic redirection patterns, which would explain the crosstalk relationship between ZIKV infection and the AdipoR–AMPK axis, must therefore be evaluated in future research. It should be added that the direct action of ZIKV in inhibiting *ADIPOR1/2* expression cannot be ruled out. Indeed, ZIKV’s ability to reduce the expression of certain genes involved in metabolic control was not entirely surprising, as shown by numerous transcriptome analyses [75,97]. Here again, the mechanisms by which ZIKV directs targeted gene silencing remain to be explored. Taken together, these observations point to the potential beneficial effect of AdipoR1/AMPK pathway stimulation during ZIKV infection (Figure 6). This can be driven by adiponectin or its pharmacological analog AdipoRon with benefits at different levels. Like in a recent study on new therapeutic opportunities to fight West Nile Virus [86], our study demonstrates the value of antiviral molecules targeting the potentiation of responses and metabolic orientation of the host cells rather than targeting the virus particle or viral factors involved in the viral cycle.

Our results suggest that the preventive rather than therapeutic use of adiponectin receptor agonists could be proposed until effective vaccines against ZIKV become available. AdipoRon would be particularly suitable for the prophylactic treatment of pregnant women. Its ease of oral use, assessed bioavailability and stability and known access to the CNS make it a particularly attractive candidate for neuroprotection in the context of ZIKV infection. AdipoRon has shown promising therapeutic potential in experimental models and preclinical studies [39,40,41,42,98]. However, finding a treatment that targets the developing fetus and can be safely administered to the pregnant mother is a daunting task. Concerning adiponectin signaling, numerous studies have been carried out to establish its role during pregnancy, even though circulating adiponectin in pregnant women appears to be compartmentalized between the maternal and fetal systems [99]. In particular, placental signaling by AdipoR1, associated with PPAR-α activation, has been shown to benefit fetal growth [100,101]. With regard to adiponectin receptor agonists, AdipoRon transfer across the placenta has not been formally verified. However, a study in rats showed that the oral administration of AdipoRon to gestating females with induced diabetes produced positive antioxidant and anti-inflammatory effects and improved the metabolic status in their offspring [102]. Further pharmaco-toxicological studies of the effects of AdipoRon on the fetus are therefore required. Clinical trials will of course be necessary to evaluate its translational therapeutic uses, but this drug repositioning approach of agonists of the adiponectin signaling pathways seems to us to hold out hope for people wishing to procreate in ZIKA-endemic areas.

Ultimately, this study holds promising implications for the development of active treatments against ZIKV. It represents a foundational step toward establishing a preclinical model for investigating the favorable effects of adiponectin receptor agonists not only on viral load but also on neurogenesis during ZIKV infection. Further progression of our research is imperative, through ex vivo exploration using brain-like cerebral organoids derived from human embryonic stem cells. This technology has already proved its relevance in studying the impact of ZIKV on brain development and the effect of drugs to combat it [103,104,105]. In addition, we need to develop in vivo investigations in mice [106], a choice model offering knockout (KO) mutations in various genes related to the AdipoR1/AMPK/CPT1a signaling pathway [107,108,109,110]. This step will be a prerequisite for moving on to clinical trials. Of note, extrapolating in vitro results to in vivo scenarios introduces several uncertainties due to the inherent differences between these two environments. Factors such as absorption, distribution, metabolism and excretion (ADME) processes, as well as pharmacokinetics, can significantly vary between in vitro and in vivo settings, affecting the bioavailability and efficacy of AdipoRon and adiponectin. Acknowledging and addressing these uncertainties is crucial for a comprehensive interpretation of in vitro findings and their potential relevance to in vivo situations.

In conclusion, we are still in the early stages of understanding how a cell’s metabolic orientation can exert a pro- or antiviral effect [111]. This opens the way to antiviral mechanisms that need to be further deciphered. Our study has the merit of identifying the antiviral potential of the AdipoR1/AMPK/PPAR-α/CPT1 signaling pathway against ZIKV. Targeting host cell responses and their metabolic orientation is therefore a promising strategy for the development of broad-spectrum antiviral drugs that can effectively combat various viruses sharing common strategies in interacting with host cells, and in particular opens up innovative prospects for the treatment of ZIKV-specific pathologies.

## Figures and Tables

**Figure 1 viruses-16-00024-f001:**
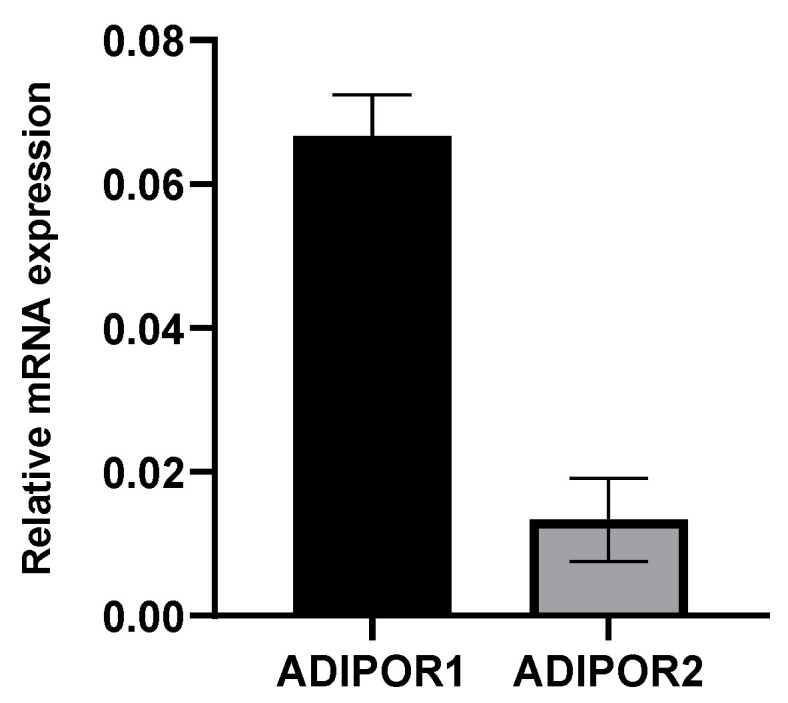
Adiponectin receptor *ADIPOR1* and *ADIPOR2* expression in A549 cell model. The basal expression level of *ADIPOR1* and *ADIPOR2* mRNA was measured using qRT-PCR in A549 cells under normal growing conditions. Expression levels were normalized to the housekeeping gene (*RPLP0*). *ADIPOR1* is almost six-fold more expressed than *ADIPOR2* in A549 cells. The error bars represent the standard deviations of three independent experiments.

**Figure 2 viruses-16-00024-f002:**
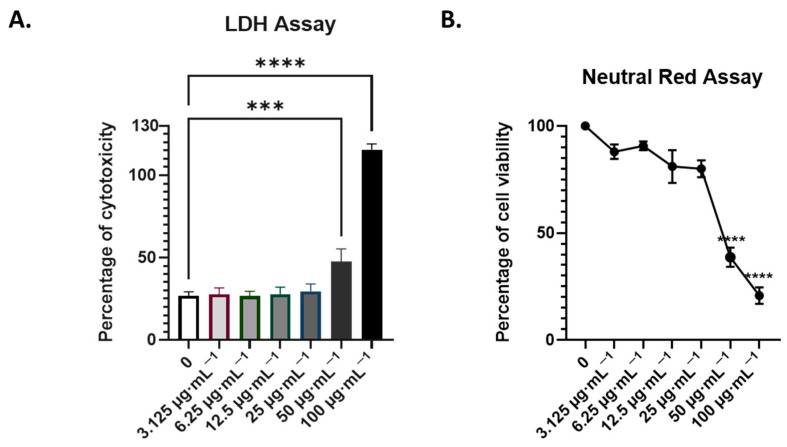
Dose-dependent cytotoxicity of AdipoRon on A549 cells. AdipoRon-related cytotoxicity was assessed using (**A**)LDH and (**B**) neutral red assays. Cells were plated and incubated for 24 h with different concentrations of AdipoRon (from 3.125 to 100 µg·mL^−1^, or 7.3 to 233 µM) as indicated in the figures. Values represent the mean and the standard deviations of three independent experiments, *** *p* < 0.0002; **** *p* < 0.0001.

**Figure 3 viruses-16-00024-f003:**
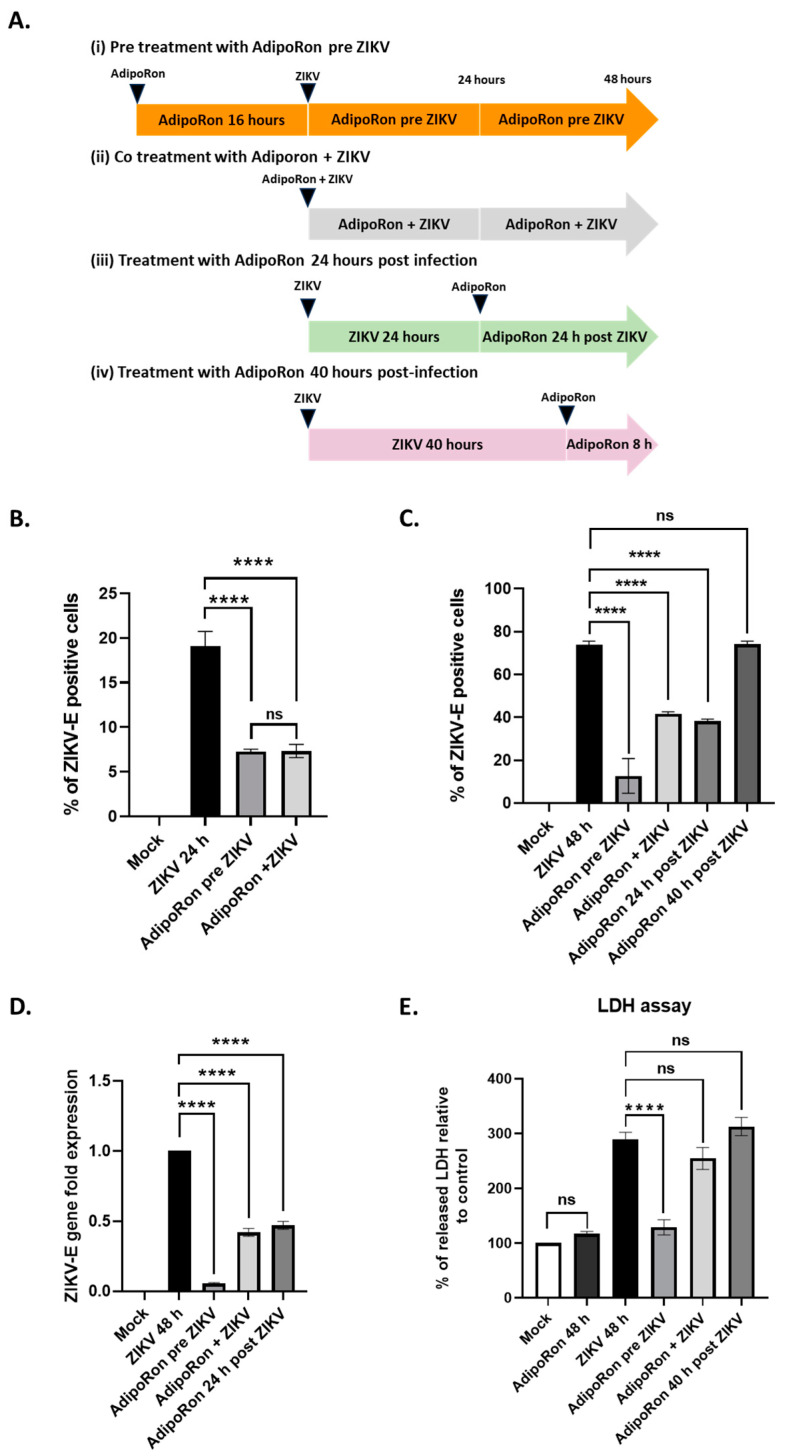
AdipoRon treatment of A549 cells protects against ZIKV infection and cytopathy. (**A**) Schematic illustration of the experimental design of AdipoRon treatments and ZIKV infection. (**B**) Percentage of ZIKV-E positive cells immunodetected using the 4G2 antibody, 24 h post-infection with ZIKV^PF13^ at MOI 2. (**C**) Percentage of ZIKV-E-positive cells immunodetected using the 4G2 antibody, 48 h post-infection with ZIKV^PF13^ at MOI 2. (**D**) ZIKV replication was determined via quantification of the viral RNA levels (*ZIKV-E*) using qRT-PCR. Expression levels were normalized to the housekeeping gene (*RNAPII*) and fold changes are related to viral RNA quantification at 48 h post-infection. (**E**) Percentage of cytotoxicity measured by LDH release in cell culture supernatants of treated and infected cells. Values represent the mean and the standard deviations of three independent experiments, ns: non-significant; **** *p* < 0.0001.

**Figure 4 viruses-16-00024-f004:**
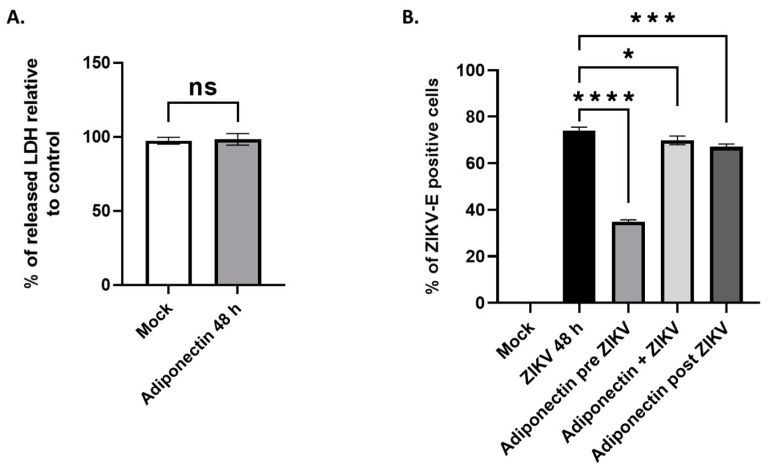
A549 cell treatment with adiponectin and ZIKV infection. (**A**) Effect of 5 µg·mL^−1^ of adiponectin treatment on cytotoxicity, related to non-treated cells (measured by LDH release in cell culture supernatants). (**B**) Effects of adiponectin treatment (5 µg·mL^−1^) on the percentage of ZIKV-E-positive cells immunodetected by the 4G2 antibody, 48 h post-infection. Values represent the mean and the standard deviations of three independent experiments. ns = not significant, * *p* < 0.05; *** *p* < 0.0002; **** *p* < 0.0001.

**Figure 5 viruses-16-00024-f005:**
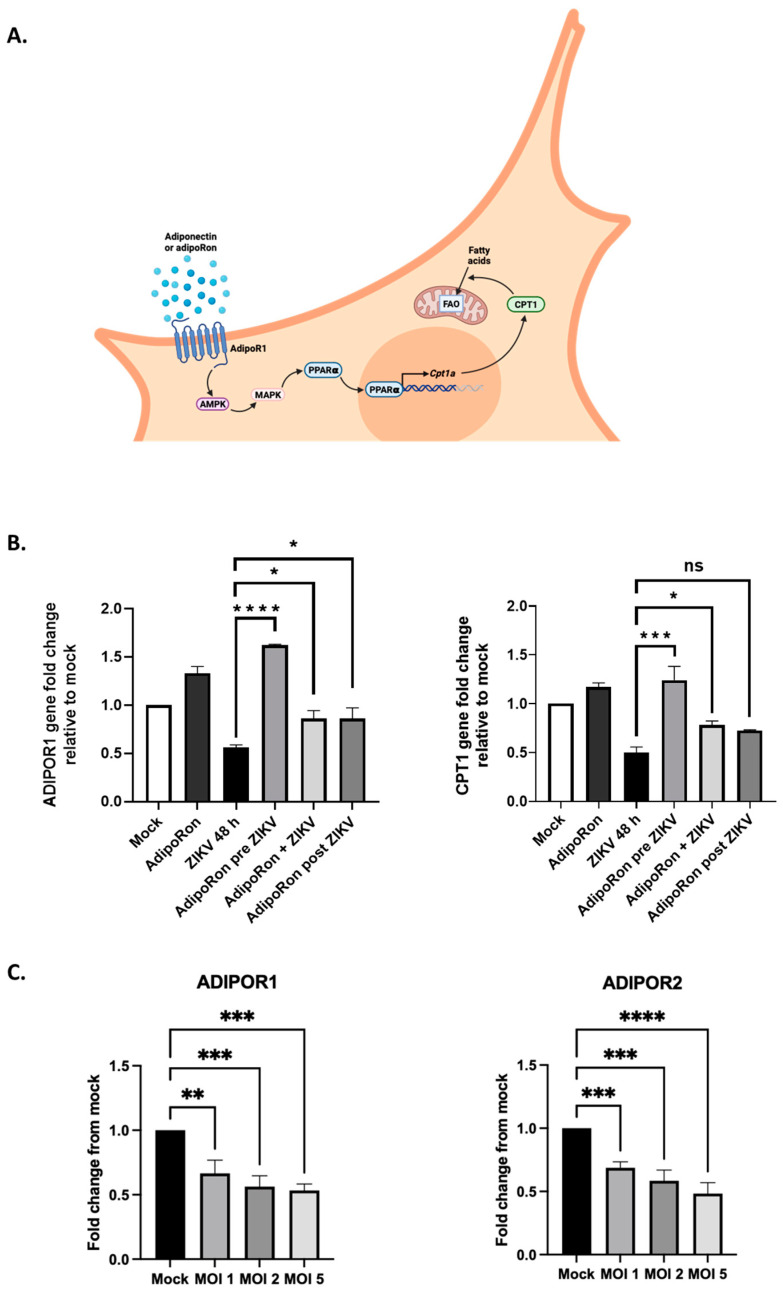
ZIKV infection and AdipoRon treatment of A549 cells modify gene expression in the adiponectin signaling pathway. (**A**) Schematic representation of the PPARα–CPT1 axis of the AdipoR1/AMPK signaling pathway. (**B**) The *ADIPOR1* gene and the *CPT1* gene, a gene under the control of the adiponectin/AdipoR1 pathway, are downregulated after 48 h of ZIKV infection. (**C**) ZIKV controls adiponectin receptor expression. Expression of the *CPT1*, *ADIPOR1* and *ADIPOR2* genes was normalized to the *RNAPII* housekeeping gene. Values represent the mean and the standard deviations of three independent experiments. ns = not significant, * *p* < 0.05; ** *p* < 0.001; *** *p* < 0.0002; **** *p* < 0.0001.

**Figure 6 viruses-16-00024-f006:**
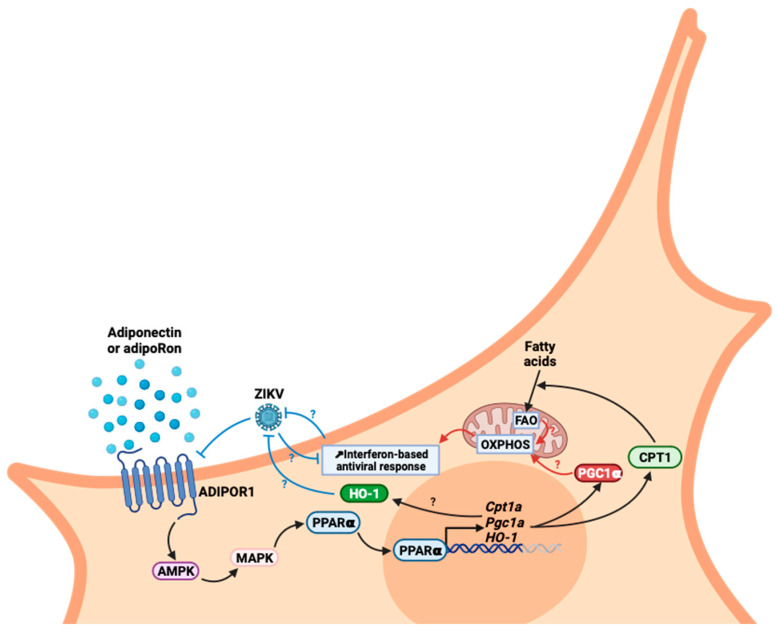
Hypotheses for antiviral modes of action of AdipoR1/AMPK signaling and its PGC-1α and CPT1 axis on infection and ZIKV’s ability to interfere with these pathways.

**Table 1 viruses-16-00024-t001:** Listing of the primers used for qRT-PCR.

Gene	Forward Primer	Reverse Primer
*GAPDH*	5′-CAAATTCCATGGCACCGTCA-3′	5′-GGAGTGGGTGTCGCTGTTGA-3′
*RNAPII*	5′-GAGAGCGTTGAGTTCCAGAACC-3′	5′-TGGATGTGTGCGTTGCTCAGCA-3′
*RPLP0*	5′-AGATGCAGCAGATCCGCAT-3′	5′-GGATGGCCTTGCGCA-3′
*ADIPOR1*	5′-AGCCTGCGGCTTAATTTGAC-3′	5′-CAACTAAGAACGGCCATGCA-3′
*ADIPOR2*	5′-GCAGCCAAGTTTTACCGAAG-3′	5′-CACCTCAAATGTGGGCTTTT-3′
*CPT1a*	5′-GATCCTGGACAATACCTCGGAG-3′	5′-CTCCACAGCATCAAGAGACTGC-3′
*ZIKV-E*	5′-CTGGTCACCTGGGGAAACTA-3′	5′-GAGCCTTCTCAAAGCACACC-3′

## Data Availability

All data is available in this manuscript.

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
