# Peer review of "The Antiviral Potential of AdipoRon, an Adiponectin Receptor Agonist, Reveals the Ability of Zika Virus to Deregulate Adiponectin Receptor Expression"

_viruses, 2023, doi:10.3390/v16010024_

Round 1
Reviewer 1 Report
Comments and Suggestions for Authors
In the manuscript "The antiviral potential of AdipoRon, an adiponectin receptor agonist, reveals the ability of Zika virus to deregulate adiponectin receptor expression" by Safadi and co-workers, the antiviral activity of AdipoRon and adiponectin against Zika virus-infected A549 cells was analyzed. Infection with ZIKV, mainly transmitted by mosquitoes of the genus Aedes, could lead to severe neurological diseases like the congenital ZIKA syndrome in infants. Neuroprotective compounds that also show antiviral activity are an interesting therapeutic option to treat ZIKV-infected patients. Adiponectin, a peptide hormone that increases insulin sensitivity, is a molecule with pleiotropic protective properties. Importantly, the adiponectin receptor agonist AdipoRon has beneficial effects on brain function and might be neuroprotective via the AdipoR1/AMPK signaling pathway. Therefore, the authors investigated the antiviral effects of pre or co-treatment with AdipoRon and the effects on adiponectin signaling in ZIKV-infected A549 cells.
The finding that activation of the adiponectin signaling pathway has anti-ZIKV capacity is novel and of importance. However, the experiments are preliminary and need to be improved before publication.
I recommend accepting the manuscript after major revision.
Major points:
1. Figure 1 at page 5: The quantification of ADIPOR1 and 2 mRNA levels is not clear. What does the numbering on the y-axis mean? How are the data normalized? Please provide the ct-values and also the delta delta ct values! Add more relevant information about qRT-PCR in the methods section (page 4, line 157) so that the reader can follow up on how the experiment was exactly done.
2. Figure 2 at page 5: Provide information about the CC50 values for both assays; CC50 values should be presented as nanomolar or micromolar concentrations.
3. Figure 3 at page 6: I strongly recommend determining the virus titers by TCID50 or Plaque Assays in addition to ZIKV-E positive cells! Figure 4 can be included in Figure 3.
4. Discussion, page 11 lines 356 to 359: It would be helpful to test if there is a shift from OXPHOS to more glycolysis in infected A549 cells and if AdipoRon might convert such a metabolic shift.
Minor points:
1. How selective is AdipoRon? Can it cross the plasma membrane and bind to cellular targets? Has anybody checked cell permeability of AdipoRon? Please comment on this!
2. Introduction, line 98: The antiviral effects of AdipoRon are only substantial under pre-treatment conditions. Delete co-treatment!
3. Materials and Methods, lines 114 and 115: How is AdipoRon dissolved as a stock solution? In DMSO? Please provide the reader with concentrations given in nanomolar or micromolar (not only µg per ml). See also line 127!
4. Results, Figure 3A page 6: Change iiii) into iv); 3B – why is the number of ZIKV-E positive cells less than 20% after 24 h?
5. Pre- and co-treatment of A549 cells with AdipoRon were done at 25 µg x ml -1. However, strong cytotoxic effects are obvious at 50 µg x ml -1. It seems that the therapeutic window is not very big for AdipoRon. Please discuss this point! The antiviral activity in the pre-treatment setup is about a 70% reduction of infected cells. This is less than one log-phase. For efficient antiviral activity, it is recommended that the log reduction should be more than one log-phase. Please discuss this point since the antiviral activity is only weak with AdipoRon. Due to the results from pre-, co-, and post-infection treatment, it seems that AdipoRon can only be used prophylactically. Add this information in the discussion section!
Comments on the Quality of English LanguageThe quality of English is fine and needs only minor editing
Author Response
In the manuscript "The antiviral potential of AdipoRon, an adiponectin receptor agonist, reveals the ability of Zika virus to deregulate adiponectin receptor expression" by Safadi and co-workers, the antiviral activity of AdipoRon and adiponectin against Zika virus-infected A549 cells was analyzed. Infection with ZIKV, mainly transmitted by mosquitoes of the genus Aedes, could lead to severe neurological diseases like the congenital ZIKA syndrome in infants. Neuroprotective compounds that also show antiviral activity are an interesting therapeutic option to treat ZIKV-infected patients. Adiponectin, a peptide hormone that increases insulin sensitivity, is a molecule with pleiotropic protective properties. Importantly, the adiponectin receptor agonist AdipoRon has beneficial effects on brain function and might be neuroprotective via the AdipoR1/AMPK signaling pathway. Therefore, the authors investigated the antiviral effects of pre or co-treatment with AdipoRon and the effects on adiponectin signaling in ZIKV-infected A549 cells.
The finding that activation of the adiponectin signaling pathway has anti-ZIKV capacity is novel and of importance. However, the experiments are preliminary and need to be improved before publication.
I recommend accepting the manuscript after major revision.
We would like to thank reviewer 2 for the careful and critical reading of our manuscript.
We have therefore made some clarifications in reply to the comments made. Our responses to the points raised are shown in red below each point. The same applies to changes made to the text of the manuscript.
Major points:
- Figure 1 at page 5: The quantification of ADIPOR1 and 2 mRNA levels is not clear. What does the numbering on the y-axis mean? How are the data normalized? Please provide the ct-values and also the delta delta ct values! Add more relevant information about qRT-PCR in the methods section (page 4, line 157) so that the reader can follow up on how the experiment was exactly done.
Thank you for raising this point that was insufficiently described.
Accordingly, we modified the material and methods section to give more information on the qRT-PCR analysis. We have also modified the legend and y-axis of Figure 1.
In response to your request, we have provided the Ct in the following table:
In Materials and Methods
now Line 197, we added:
The qPCR data sets were analyzed using the ΔΔCt method (Livak and Schmittgen, 2001) and the results were normalized to the values obtained for the housekeeping genes.
- Figure 2 at page 5: Provide information about the CC50 values for both assays; CC50 values should be presented as nanomolar or micromolar concentrations.
The neutral red assay is considered a reliable method for estimating cytotoxicity, with high sensitivity compared to LDH measurements (In vitro cytotoxicity assays: Comparison of LDH, neutral red, MTT and protein assay in hepatoma cell lines following exposure to cadmium chloride. George Fotakis, John A. Timbrell (https://doi.org/10.1016/j.toxlet.2005.07.001). Therefore, the concentration of AdipoRon that reduces cell viability by 50% was estimated from the neutral red assay curve showed below.
Upon request, we have also converted concentration to molarity. Adiporon has a molecular weight of 428.5g/mol. The CC50 value of 55.06 µg.mL-1 or 128,5 µM has been added to the result section. The concentration of 25 µg.mL-1 of AdipoRon used to treat cells corresponds to 58µM.
line 243 : The 50% cytotoxic concentration (CC50) was calculated from the dose-response curve of viability obtained using the neutral red assay (Figure 2B). CC50 is estimated at 55 µg·mL-1 corresponding to 128,5µM. We then considered 25 µg·mL-1 which corresponds to 58µM of AdipoRon to be the highest non-cytotoxic concentration we could assess on A549 cells.
In addition, as requested by the other reviewers, we have added to the introduction and discussion with more data on AdipoRon and its already well-documented use in experimental models and preclinical studies. We draw your attention to the fact that many studies reporting the use of adiporon for in vitro experiments or dose administration in animal models frequently use µg.ml-1, or of course µg.kg-1 for oral or injection administration.
For exemple :
in vitro use in different osteosarcoma cell lines (Saos-2 and U2OS) (Sapio et al. AdipoRon Affects Cell Cycle Progression and Inhibits Proliferation in Human Osteosarcoma Cells. Journal of oncology doi: 10.1155/2020/7262479,)
and other examples in the following articles: doi: 10.3390/biomedicines11092585, https://doi.org/10.1016/j.psj.2022.102319, https://doi.org/10.3390/ijms22147516
For oral administration, or intraperitoneal injection.
( A small-molecule AdipoR agonist for type 2 diabetes and short life in obesity, Nature. 2013, Okada-Iwabu et al. doi: 10.1038/nature12656)(AdipoRon Attenuates Neuroinflammation After Intracerebral Hemorrhage Through AdipoR1-AMPK Pathway. Neuroscience. 2019. doi: 10.1016/j.neuroscience.2019.05.060)
- Figure 3 at page 6: I strongly recommend determining the virus titers by TCID50 or Plaque Assays in addition to ZIKV-E positive cells! Figure 4 can be included in Figure 3.
We are well aware that it would be a plus to provide pfu titrations of viral progenies in the culture supernatants following the treatments carried out. Nevertheless, we believe that the information we provide with the relative quantification of intracellular viral RNAs (Figure 3D) is a convincing tool of estimating viral replication and inhibitory effects on this replication. Pre-treatment of cells with adipoRon leads to 70% to 85% reduction in infectivity and a 20-fold reduction in the detectable viral genome, attesting to an antiviral activity. A plant extract treatment of ZIKV-infected cells has demonstrated a reduction in the infection to two-thirds measured by flow cytometry. This decrease corresponded to a 2-log reduction in the viral titer. (doi:10.1038/s41598-018-29183-2). Others studies as well showed the same trend (doi: 10.3390/ijms20102382, doi: 10.3390/molecules25102316, doi: 10.3390/microorganisms8091257, doi: 10.3390/molecules24193447.). Collectively, these studies demonstrate that reductions in infectivity of the same order of magnitude as those we found, are associated with a 2-log reduction, which is considered sufficient to validate antiviral potential. AdipoRon is therefore well within the range of characteristics expected for an antiviral agent of interest.
- Discussion, page 11 lines 356 to 359: It would be helpful to test if there is a shift from OXPHOS to more glycolysis in infected A549 cells and if AdipoRon might convert such a metabolic shift.
You raise a point that would obviously be very interesting to decipher. It seems necessary to explore the impact of metabolic reprogramming on infection, as this would offer new cellular targets for antiviral research. We are only at the beginning of understanding these mechanisms, and the literature review shows often ambiguous results on the pro- or antiviral effects exerted by the metabolic shift from oxphos to aerobic glycolysis. Further studies, using oximetry measurements for example, are needed on the infection of cells in vitro, but also on the stimulation of the AdipoR1/AMPK/CPT1 pathway by adipoR receptor agonists.
We have added these perspectives in the discussion section, line 527.
Minor points:
- How selective is AdipoRon? Can it cross the plasma membrane and bind to cellular targets? Has anybody checked cell permeability of AdipoRon? Please comment on this!
As previously mentioned, AdipoRon is a molecule that has already been tested for use in a number of contexts, notably pre-clinical. We have added a number of data and references on the properties of this molecule in the introductory sections and in the discussion. This molecule acts on the cell surface, at membrane level, as a ligand for adipoR receptors and induces signal transduction. To our knowledge, it does not diffuse into cells.
- Introduction, line 98: The antiviral effects of AdipoRon are only substantial under pre-treatment conditions. Delete co-treatment!
We modified it accordingly, now line 113.
- Materials and Methods, lines 114 and 115: How is AdipoRon dissolved as a stock solution? In DMSO? Please provide the reader with concentrations given in nanomolar or micromolar (not only µg per ml). See also line 127!
Many thanks for your careful reading, we have corrected this omission.
Adiporon was dissolved in ethanol (line 129 : AdipoRon was dissolved in absolute ethanol and adiponectin in sterile water as recommended by manufacturers.) and the corresponding nanomolar concentrations were added as requested (line 136, 245 and 255).
- Results, Figure 3A page 6: Change iiii) into iv);
This was done.
3B – why is the number of ZIKV-E positive cells less than 20% after 24 h?
Characterization of in vitro infection of A549 cells by ZIKV clinical isolate PF-25013-18 has been the subject of several publications, notably by our group (doi 10.1016/j.virol.2016.03.006). The % of 4G2 positive cells during the infection kinetics depends on the MOI. Here we were at an MOI of 2, and the average amplitude of infection, with a rate of 20% at 24 h and 70% at 48h are in the same range as the data already obtained.
- Pre- and co-treatment of A549 cells with AdipoRon were done at 25 µg x ml -1. However, strong cytotoxic effects are obvious at 50 µg x ml -1. It seems that the therapeutic window is not very big for AdipoRon.
Yes, we concur that AdipoRon appears to exhibit a narrow therapeutic window, as evidenced by a significant decrease in cell viability observed between the concentrations of 25 and 50 µg/ml. Further in vitro and in vivo testing of AdipoRon concentrations is required. If a similarly narrow window is confirmed, it underscores the importance of exercising vigilance to determine and utilize the appropriate concentration.
As mentioned above, the main advantage of adipoRon is related to a drug repositioning strategy of a molecule with established bioavailability and usable concentrations in vivo. Adverse Adverse Adverse We have taken your comment into consideration and discussed it further line 491 to 507
Please discuss this point! The antiviral activity in the pre-treatment setup is about a 70% reduction of infected cells. This is less than one log-phase. For efficient antiviral activity, it is recommended that the log reduction should be more than one log-phase. Please discuss this point since the antiviral activity is only weak with AdipoRon.
Thank you for your comment. We have provided some insights regarding this appraisal of what is a valuable antiviral in point 3.
Due to the results from pre-, co-, and post-infection treatment, it seems that AdipoRon can only be used prophylactically. Add this information in the discussion section!
We addressed this point in the discussion line 489.

Reviewer 2 Report
Comments and Suggestions for Authors
1. While the introduction touches on the uniqueness of Zika virus in its ability to cause severe neurological diseases and the lack of specific treatments, consider providing a more explicit link between these characteristics and the rationale for exploring AdipoRon. Elaborate on why AdipoRon, as an adiponectin receptor agonist, is a relevant candidate for investigation in the context of Zika virus-induced neurological damage. Specify if there are unique aspects of AdipoRon's mechanism of action that make it particularly suitable for addressing the neurological impact of Zika virus.
2. Briefly expand on the choice of A549 epithelial cells as the model system for the in vitro study. Justify why these cells were selected, considering their relevance to Zika virus infection and their ability to express adiponectin receptors. Discuss any advantages or limitations of this cell model and how it aligns with the objectives of the study. Providing more context on the rationale behind the chosen experimental model will enhance the overall scientific rigor.
3. Emphasize the potential clinical implications of the study findings. Discuss how the identification of AdipoRon's antiviral effects on Zika virus-infected cells could pave the way for therapeutic interventions or preventive strategies. Consider discussing the broader implications for pregnant women or those planning to give birth in Zika epidemic and endemic zones. This will underscore the translational significance of the research.
4. Given the unexpected finding that ZIKV downregulates ADIPOR1 and CPT1 gene expression, suggest additional experiments to elucidate the underlying mechanisms. For example, investigating the impact of ZIKV infection on key signaling pathways involved in adiponectin receptor regulation could provide insights. Consider experiments such as Western blotting or immunofluorescence to assess protein levels and localization. Additionally, exploring the direct interaction between ZIKV components and adiponectin receptor genes at the molecular level could contribute to a more comprehensive understanding of the observed effects.
5. Elaborate on the downstream signaling pathways influenced by AdipoRon and ZIKV. Since the study mentions the involvement of AMPK and PPARα in the AdipoR1 signaling pathway, consider measuring their activation states or downstream target gene expression levels. This could provide mechanistic insights into how AdipoRon and ZIKV modulate the broader adiponectin signaling network. Understanding the cascade of molecular events will contribute to a more detailed and interconnected interpretation of the results.
6. Explore the cellular localization of AdipoR1 and CPT1 in response to AdipoRon treatment and ZIKV infection. Immunofluorescence or immunohistochemistry experiments can reveal whether changes in gene expression translate into alterations in protein localization within the cell. This information is crucial for understanding the functional consequences of AdipoRon and ZIKV on the subcellular dynamics of AdipoR1 and CPT1.
7. Complement gene expression data with functional assays to assess the activity of the AdipoR1/AMPK pathway. For instance, measuring AMPK phosphorylation levels or assessing the transcriptional activity of PPARα could provide direct evidence of pathway activation or inhibition. Integrating functional assays will strengthen the biological relevance of the observed gene expression changes and further support the study's conclusions.
8. Acknowledge any limitations of the study and propose avenues for future research. For instance, discuss potential challenges or uncertainties in extrapolating in vitro findings to in vivo scenarios, and suggest experimental approaches to validate the observed effects in animal models or ex-vivo models of developing brains. Highlight the need for further investigation into the mechanisms underlying the observed interactions between ZIKV and the AdipoR1/AMPK pathway.
9. Integrate the current findings with existing literature on Zika virus pathogenesis and the development of antiviral strategies. Discuss how the AdipoR1/AMPK pathway's antiviral potential aligns or contrasts with other reported antiviral mechanisms. This can provide a broader context for readers and underscore the significance of the current study within the larger scientific landscape.
10. Emphasize the novelty of the study and its contribution to the field. Clearly articulate how the identification of the AdipoR1/AMPK pathway's antiviral potential against ZIKV adds new insights to existing knowledge. Highlight the significance of targeting host cell responses and metabolic orientation as a promising avenue for antiviral drug development.
Comments on the Quality of English LanguageModerate editing of English language required
Author Response
We thank the reviewer for the careful reading of our manuscript and helpful suggestions and critiques. They have enabled us to improve and clarify our message, and to accentuate the importance of the current study within a broader scientific landscape.
We have addressed all of Reviewer 1's suggestions, which are indicated point-by-point below. Our responses are in red below each point. The same applies to changes made to the text in the manuscript.
Comments and Suggestions for Authors
- While the introduction touches on the uniqueness of Zika virus in its ability to cause severe neurological diseases and the lack of specific treatments, consider providing a more explicit link between these characteristics and the rationale for exploring AdipoRon. Elaborate on why AdipoRon, as an adiponectin receptor agonist, is a relevant candidate for investigation in the context of Zika virus-induced neurological damage. Specify if there are unique aspects of AdipoRon's mechanism of action that make it particularly suitable for addressing the neurological impact of Zika virus.
We agree that our introduction insufficiently describes the rationale for choosing AdipoRon to study its potential benefits on the brain pathophysiology induced by ZIKV infection. We have extensively revised the introduction to more fully discuss the main advantages of this adiponectin receptor agonist, providing additional information and associated references on its pharmacological characteristics. We have also added some details in the discussion section on its pharmacological interest in treating various pathologies, including neurological pathologies dependent on the AdipoR1/AMPK pathway.
line 60
In this context, the search for a molecule with a neuroprotective and antiviral role adapted to this particular situation of pregnancy becomes a priority.
line 67
Adiponectin signaling depends on two main types of adiponectin receptors, AdipoR1 and AdipoR2, which are ubiquitously expressed in a wide range of human tissues [12] + (Adiponectin receptors: a review of their structure, function and how they work.Yamauchi T, Iwabu M, Okada-Iwabu M, Kadowaki T. Best Pract Res Clin Endocrinol Metab. 2014 Jan;28(1):15-23. doi: 10.1016/j.beem.2013.09.003. )
line 86 :
AdipoRon is a small synthetic molecule that was the first orally active adiponectin receptor agonist to be discovered (Okada-Iwabu et al. "A small-molecule AdipoR agonist for type 2 diabetes and short life in obesity". Nature. doi:10.1038/nature12656. Holland WL ; Scherer PE "Cell Biology. Ronning after the adiponectin receptors". Science. 342 (6165): 1460–1461.doi:10.1126/science.1249077.). Several studies have shown its beneficial effects on brain functions and neurological disorders [28]+ 50(doi:10.1038/s41380-020-0701-0.). AdipoRon offers a number of pharmacological advantages linked to its ability to cross the blood-brain barrier. It has been detected in the brain both following oral administration and after systemic injection (50=doi:10.1038/s41380-020-0701-0. + Nicolas et al. “Adiporon, an adiponectin receptor agonist acts as an antidepressant and metabolic regulator in a mouse model of depression.” Transl Psychiatry 8, 159 (2018). doi:10.1038/s41398-018-0210-y). A low dose of AdipoRon promoted hippocampal cell proliferation and increased serum adiponectin levels in mice[29]. Indeed, modulation of adiponectin signaling (whether by injection of adiponectin or AdipoRon) enhanced neuronal cell survival in various in vivo and in vitro models, including cerebral ischemia, cerebral hemorrhage and oxygen and glucose deprivation experiments. (Hao Bai et al. 2018 “Adiponectin confers neuroprotection against cerebral ischemia-reperfusion injury through activating the cAMP/PKA-CREB-BDNF signaling” DOI: 10.1016/j.brainresbull.2018.10.013 ; Xun Wu et al. “Recombinant Adiponectin Peptide Ameliorates Brain Injury Following Intracerebral Hemorrhage by Suppressing Astrocyte-Derived Inflammation via the Inhibition of Drp1-Mediated Mitochondrial Fission” DOI: 10.1007/s12975-019-00768-x ; Jun Yu et al. “AdipoRon Protects Against Secondary Brain Injury After Intracerebral Hemorrhage via Alleviating Mitochondrial Dysfunction: Possible Involvement of AdipoR1-AMPK-PGC1α Pathway” Neurochem Res. 2019 DOI: 10.1007/s11064-019-02794-5 ; Shenghao Zhang et al. “Adiponectin/AdiopR1 signaling prevents mitochondrial dysfunction and oxidative injury after traumatic brain injury in a SIRT3 dependent manner” Redox Biol. 2022 DOI: 10.1016/j.redox.2022.102390 ; Bodong Wang et al. “Adiponectin Attenuates Oxygen-Glucose Deprivation-Induced Mitochondrial Oxidative Injury and Apoptosis in Hippocampal HT22 Cells via the JAK2/STAT3 Pathway.” Cell Transplant.2018. DOI: 10.1177/0963689718779364).
The results include effective neuroprotection, reduced infarct size and improved neurological outcomes.
Line 96
In addition, AdipoRon has been shown to compensate for the depletion of neural stem cells in the hippocampus by activating the AdipoR1/AMP-activated protein kinase (AMPK) pathway[20].
Based on the deleterious impact of ZIKV on neural progenitor cell proliferation and differentiation, we hypothesized that targeting adiponectin signaling could be an interesting therapeutic avenue for limiting ZIKV-induced damages in fetuses Therefore, AdipoRon, which might provide a beneficial neuroprotective effect during ZIKV infection, emerged as a relevant, easy-to-use candidate, with already tried-and-tested therapeutic delivery options and efficacy (Okada-Iwabu et al. “Perspective of Small-Molecule AdipoR Agonist for Type 2 Diabetes and Short Life in Obesity” Diabetes Metab J. 2015 doi: 10.4093/dmj.2015.39.5.363 ; Zatorski et al. “AdipoRon, an Orally Active, Synthetic Agonist of AdipoR1 and AdipoR2 Receptors Has Gastroprotective Effect in Experimentally Induced Gastric Ulcers in Mice” Molecules. 2021 doi: 10.3390/molecules26102946 ; Kim and Whee Park “Mechanisms of Adiponectin Action: Implication of Adiponectin Receptor Agonism in Diabetic Kidney Disease” Int J Mol Sci. 2019 doi:10.3390/ijms20071782. ; Khandelwal et al. “AdipoRon induces AMPK activation and ameliorates Alzheimer's like pathologies and associated cognitive impairment in APP/PS1 mice” Neurobiol Dis. 2022 doi: 10.1016/j.nbd.2022.105876.).
Line 380:
This is why we considered adiponectin, and more specifically its pharmacological analog AdipoRon, as a potential candidate of interest. These molecules, already known to exert neuroprotective activity fit the essential criteria set out above[15,32]. Additionally, AMPK dysregulation and metabolic dysfunction are found in neurodegenerative processes, and activation of the AdipoR1/AMPK pathway by AdipoRon has demonstrated therapeutic benefits in neuroinflammation following intracerebral hemorrhage. (Zheng et l. AdipoRon Attenuates Neuroinflammation After Intracerebral Hemorrhage Through AdipoR1-AMPK Pathway. Neuroscience. 2019 doi: 10.1016/j.neuroscience.2019.05.060.; Yu et al. AdipoRon Protects Against Secondary Brain Injury After Intracerebral Hemorrhage via Alleviating Mitochondrial Dysfunction: Possible Involvement of AdipoR1-AMPK-PGC1α Pathway. Neurochem Res. 2019 doi: 10.1007/s11064-019-02794-5.).
- Briefly expand on the choice of A549 epithelial cells as the model system for the in vitro study. Justify why these cells were selected, considering their relevance to Zika virus infection and their ability to express adiponectin receptors. Discuss any advantages or limitations of this cell model and how it aligns with the objectives of the study. Providing more context on the rationale behind the chosen experimental model will enhance the overall scientific rigor.
As suggested, we have completed the rationale for choosing A549 cells for our study. We have added these elements to the introduction and foreword to the results section.
Line 104
To conduct an in vitro study of the effect of AdipoRon treatment on ZIKV infection, we used the A549 epithelial cells. This cell model had the advantage of already being well characterized for its permissiveness to ZIKV (30), and was then widely validated for in vitro testing of pharmaceuticals and antiviral compounds. (Oeyen et al. “In-Depth Characterization of Zika Virus Inhibitors Using Cell-Based Electrical Impedance. Microbiol Spectr. 2022 doi: 10.1128/spectrum.00491-22. ; Morales Vasquez et al. “Identification of Inhibitors of ZIKV Replication” Viruses. 2020 doi: 10.3390/v12091041. Santos Pereira et al. “Natural Products and Derivatives as Potential Zika virus Inhibitors: A Comprehensive Review” Viruses 2023 DOI: 10.3390/v15051211)
Line 211
A549 cells are an epithelial cell model that has been one of the most widely used to understand flaviviruses infection processes. Our team and others have characterized the in vitro infection of A549 cells by ZIKV( Frumence et al. Viruses 2016 “The South Pacific epidemic strain of Zika virus replicates efficiently in human epithelial A549 cells leading to IFN-β production and apoptosis induction” doi: 10.1016/j.virol.2016.03.006., Turpin et al. Biochimie Zika virus subversion of chaperone GRP78/BiP expression in A549 cells during UPR activation doi: 10.1016/j.biochi.2020.05.011., Gobillot et al. The Robust Restriction of Zika Virus by Type-I Interferon in A549 Cells Varies by Viral Lineage and Is Not Determined by IFITM3, viruses 2020 doi: 10.3390/v12050503.). This model has been used extensively to decipher cellular responses to infection, and in particular to assess the cytopathic effects induced by ZIKV (doi: 10.3390/cells8111338. doi: 10.3390/ijms22073750.). It is particularly suitable and widely used for testing the in vitro effects of antivirals (doi: 10.1016/j.virusres.2017.11.003. DOI: 10.3390/v15051211). As literature data on adiponectin receptor expression by A549 cells is patchy (DOI: 10.4049/jimmunol.182.1.684), we first checked whether these cells indeed expressed adiponectin receptor, to validate our cellular model for studying the effect of adipoRon on ZIKV in vitro infection.
- Emphasize the potential clinical implications of the study findings. Discuss how the identification of AdipoRon's antiviral effects on Zika virus-infected cells could pave the way for therapeutic interventions or preventive strategies. Consider discussing the broader implications for pregnant women or those planning to give birth in Zika epidemic and endemic zones. This will underscore the translational significance of the research.
We appreciate your insightful suggestion.
We emphasized this point in the discussion to highlight how the antiviral effect of AdipoRon presents a promising avenue for preventive and therapeutic strategies, more specifically for pregnant women or those planning to give birth in Zika epidemic and endemic zones.
Line 368
Among the huge efforts to find suitable prophylactics and treatments, drug repositioning offers several undeniable advantages. Proven safety and knowledge of the best drug delivery options are major strengths for the particular situation of ZIKV exposure on maternal-fetal health (Drugs to limit Zika virus infection and implication for maternal-fetal health.Front Virol. 2022;2:928599. doi: 10.3389/fviro.2022.928599.)
Line 488
Our results suggest that preventive rather than therapeutic use of adiponectin receptor agonists could be proposed until effective vaccines against ZIKV become available. AdipoRon would be particularly suitable for the prophylactic treatment of pregnant women. Its ease of oral use, assessed bioavailability and stability, and known access to the CNS make it a particularly attractive candidate for a neuroprotection in the context of ZIKV infection. AdipoRon has shown promising therapeutic potential in experimental models and preclinical studies (Barbalho et al. “AdipoRon and ADP355, adiponectin receptor agonists, in Metabolic-associated Fatty Liver Disease (MAFLD) and Nonalcoholic Steatohepatitis (NASH): A systematic review” Biochem Pharmacol. 2023 doi: 10.1016/j.bcp.2023.115871.
; Okada-Iwabu et al. “Perspective of Small-Molecule AdipoR Agonist for Type 2 Diabetes and Short Life in Obesity” Diabetes Metab J. 2015 doi: 10.4093/dmj.2015.39.5.363 ; Zatorski et al. doi: 10.3390/molecules26102946. doi: 10.3390/ijms20071782. doi: 10.1016/j.nbd.2022.105876.). However, finding a treatment that targets the developing fetus and can be safely administered to the pregnant mother is a daunting task. Concerning adiponectin signaling, numerous studies have been carried out to establish its role in the adaptive metabolic response to pregnancy and its effects on fetal development, even though circulating adiponectin in pregnant women appears to be compartmentalized between the maternal and fetal systems (for a review: Moyce Gruber and Dolinsky « The Role of Adiponectin during Pregnancy and Gestational Diabetes. » Life (Basel). 2023 doi: 10.3390/life13020301). In particular, placental signaling by AdipoR1, associated with PPAR-α activation, has been shown to benefit fetal growth with an impact on nutrient transport. (doi: 10.2337/db09-0824.). In addition, an increase in fetal adiponectin during gestation also promotes fetal growth (doi: 10.2337/db12-0055). With regard to adiponectin receptor agonists, AdipoRon transfer across the placenta has not been formally verified. However, a study in rats showed that oral administration of AdipoRon to gestating females with induced diabetes, produced positive antioxidant and anti-inflammatory effects and improved metabolic status in their offspring (Gázquez et al. Adiponectin agonist treatment in diabetic pregnant rats. J Endocrinol. doi: 10.1530/JOE-20-0617.). Further pharmaco-toxicological studies of the effects of AdipoRon on the fetus are therefore required. Clinical trials will of course be necessary to evaluate translational therapeutic uses, but this drug repositioning approach of agonists of the adiponectin signaling pathways seems to us to hold out hope for people wishing to procreate in ZIKA-endemic areas.
- Given the unexpected finding that ZIKV downregulates ADIPOR1 and CPT1 gene expression, suggest additional experiments to elucidate the underlying mechanisms. For example, investigating the impact of ZIKV infection on key signaling pathways involved in adiponectin receptor regulation could provide insights. Consider experiments such as Western blotting or immunofluorescence to assess protein levels and localization. Additionally, exploring the direct interaction between ZIKV components and adiponectin receptor genes at the molecular level could contribute to a more comprehensive understanding of the observed effects.
Thank you for pointing this out.
We fully agree that the finding that ZIKV down-regulates the expression of the ADIPOR1 and CPT1 genes opens up many mechanistic questions. In the discussion we raise the hypothesis of a metabolic antagonism with ZIKV favoring aerobic glycolysis when AdipoR1/R2 signaling enhances OXPHOS.
Of course we intend to explore the underlying mechanisms of these regulations, but we need a number of tools to carry out such a complex study. This includes antibodies and the ability to look at phosphorylation of signaling pathway targets. As you kindly suggest, we also have to correlate the observation of transcriptional regulation with actual protein levels. And we have to look if the adiponectin receptors localization is altered by ZIKV, with their sequestration in the endosome to be considered.
The question of which particular viral factor is involved in the down-regulation of adipoR expression is obviously of interest. We can take advantage of the systems we've developed in the laboratory, which enable us to express the various viral proteins separately. These experimental perspectives represent a huge amount of work to be pursued, which will undoubtedly shed light on the subject for the scientific community in a future publication.
In response to your suggestions, we have added, in the discussion part of the manuscript, some experimental approaches to address several of these points.
Line 401
To our knowledge, we have shown for the first time that the adiponectin signaling pathway has antiviral potential, making it a target that ZIKV subverts to limit its effect. The mechanisms underlying the deregulation of ADIPOR1 and CPT1 transcriptional expression by ZIKV deserve to be further investigated. Literature data on the regulation of AdipoR expression mainly mention the controls exerted by metabolic status and insulin (doi: 10.1016/j.beem.2013.09.003.). Insulin can decrease in vitro ADIPOR1/R2 expression via the phosphoinositide 3-kinase/Foxo1 pathway (DOI: 10.1074/jbc.M402367200), and AdipoR1/R2 levels are significantly decreased in muscle and adipose tissue of insulin-resistant ob/ob mice, due to hyperinsulinemia (doi.org/10.1074/jbc.M402367200). The role of metabolic reprogramming induced during ZIKV infection on AdipoR1/R2 expression then needs to be clarified. Direct activity of viral proteins cannot be ruled out. It would therefore be interesting to investigate whether viral factors would be more likely to lead to transcriptional repression (doi: 10.1038/emi.2017.9).
- Elaborate on the downstream signaling pathways influenced by AdipoRon and ZIKV. Since the study mentions the involvement of AMPK and PPARα in the AdipoR1 signaling pathway, consider measuring their activation states or downstream target gene expression levels. This could provide mechanistic insights into how AdipoRon and ZIKV modulate the broader adiponectin signaling network. Understanding the cascade of molecular events will contribute to a more detailed and interconnected interpretation of the results.
Research carried outside the context of viral infection has revealed the connection among different effectors associated with the AdipoR1/AMPK/PPARα pathway. While measuring the expression of specific genes related to this pathway and the activation states of the different targets is crucial and could offer insights into mechanistic perspectives, technical limitations prevented us from obtaining these measurements. However, we have provided additional clarification on this aspect in the results and discussion section.
Line 314
In this axis, AMPK leads to activation of p38 mitogen-activated protein kinase (MAPK) which in turn activates the peroxisome proliferator-activated receptor (PPARα). PPARα is a transcription factor that up-regulates the expression of genes associated with peroxisomal and mitochondrial β-oxidation, including the carnitine palmitoyltransferase 1 (CPT1) gene. Indeed, it has been previously demonstrated that adiponectin, by sequentially activating AMPK and p38 MAPK, increases PPARα activity in muscle cells (10.2337/db05-1322).
Line 391
While treatment of cells with AdipoRon leads to overexpression of the gene encoding AdipoR1 and that of CPT1, a transcriptional target of PPARα activated during adiponectin signaling, we revealed that ZIKV was able to interfere with this signaling pathway (Figure 5). Building upon prior studies (10.2337/db05-1322), the observed downregulation of CPT1 indicates that a potential regulation of AMPK activity, or of one or more of its downstream targets in the adiponectin/AdipoR pathway, may have occurred during infection. Experiments to measure the level of expression or activation of these factors should help to determine the effect of ZIKV on this signaling pathway.
- Explore the cellular localization of AdipoR1 and CPT1 in response to AdipoRon treatment and ZIKV infection. Immunofluorescence or immunohistochemistry experiments can reveal whether changes in gene expression translate into alterations in protein localization within the cell. This information is crucial for understanding the functional consequences of AdipoRon and ZIKV on the subcellular dynamics of AdipoR1 and CPT1.
AdipoR1 as a transmembrane receptor is localized at the plasma membrane while CPT1 resides in the outer mitochondrial membrane ( https://doi.org/10.1111/jne.13234).
An alteration in the localization of these proteins has never been described before but we agree that this possibility deserves to be investigated. As stated in our response to point 4, there are a number of issues that need to be addressed, such as the infection effect on the proteins of the adiponectin/adipoRon signaling pathways and their cellular localization. We will try to add to these interesting proposals in the future.
- Complement gene expression data with functional assays to assess the activity of the AdipoR1/AMPK pathway. For instance, measuring AMPK phosphorylation levels or assessing the transcriptional activity of PPARα could provide direct evidence of pathway activation or inhibition. Integrating functional assays will strengthen the biological relevance of the observed gene expression changes and further support the study's conclusions.
Thank you for your suggestion.
While acknowledging the potential value of such trials, as pointed above (point 5 and line306), we currently cannot pursue them due to a lack of the requisite tools at this time.
PPARα was shown to increase mRNA expression of Acyl-CoA oxidase (ACO) and carnitine palmitoyltransferase 1 (CPT1), with its activation mediated by AMPK and MAPK (doi: 10.1186/s13578-021-00587-4). Reduced expression of AdipoR1 and CPT1a (what we have shown in our study) indicates a concurrent decrease in AMPK and PPARα. Adiponectin, along with AMPK/p38 MAPK inhibitors, has been demonstrated to suppress PPARα activity (doi:10.2337/db05-1322). We believe that our study provides insight into the dynamics of the AdipoR1/AMPK pathway and that other approaches should be investigated in the future.
- Acknowledge any limitations of the study and propose avenues for future research. For instance, discuss potential challenges or uncertainties in extrapolating in vitro findings to in vivo scenarios, and suggest experimental approaches to validate the observed effects in animal models or ex-vivo models of developing brains. Highlight the need for further investigation into the mechanisms underlying the observed interactions between ZIKV and the AdipoR1/AMPK pathway.
We have taken your remarks into consideration and added a new part to the discussion.
Line 509
At the end, this study holds promising implications for the development of active treatments against ZIKV, particularly crucial for protecting pregnant women and their fetuses. It represents a foundational step towards establishing a preclinical model for investigating the favorable effects of adiponectin receptor agonists not only on viral load but also on neurogenesis during ZIKV infection. Further progression of our research is imperative, through ex-vivo exploration using brain-like cerebral organoids derived from human embryonic stem cells. This technology has already proved its relevance in studying the impact of ZIKV on brain development and the effect of drugs to combat it (Using brain organoids to understand Zika virus-induced microcephaly. Qian et al .2017 doi: 10.1242/dev.140707.; Developing human pluripotent stem cell-based cerebral organoids with a controllable microglia ratio for modeling brain development and pathology.
Xu et al. Stem Cell Reports. 2021 doi: 10.1016/j.stemcr.2021.06.011 ; Self-Organized Cerebral Organoids with Human-Specific Features Predict Effective Drugs to Combat Zika Virus Infection. Watanabe et al. Cell Rep. 2017 doi: 10.1016/j.celrep.2017.09.047.). In addition, we need to develop in vivo investigations in mice (Mouse models of Zika virus transplacental transmission. (Li et al., Antiviral Res. 2023 doi: 10.1016/j.antiviral.2022.105500.) a choice model offering knockout (KO) mutations in various genes related to the AdipoR1/AMPK/CPT1a signaling pathway (DOI: 10.1172/jci.insight.156301 ; DOI:10.2337/db06-1432 ; doi: 10.1016/j.celrep.2020.108092 ; doi: 10.1038/s42255-023-00835-6). This step will be a prerequisite for moving on to clinical trials. Of note, extrapolating in vitro results to in vivo scenarios introduces several uncertainties due to the inherent differences between these two environments. Factors such as absorption, distribution, metabolism, and excretion (ADME) processes, as well as pharmacokinetics, can significantly vary between in vitro and in vivo settings, affecting the bioavailability and efficacy of adipoRon and adiponectin. Acknowledging and addressing these uncertainties is crucial for a comprehensive interpretation of in vitro findings and their potential relevance to in vivo situations.
- Integrate the current findings with existing literature on Zika virus pathogenesis and the development of antiviral strategies. Discuss how the AdipoR1/AMPK pathway's antiviral potential aligns or contrasts with other reported antiviral mechanisms. This can provide a broader context for readers and underscore the significance of the current study within the larger scientific landscape.
We have developed these points in the discussion by including references related to antivirals against ZIKV and in the part devoted to strategies targeting host cell responses. We believe we have adequately developed the hypothesis of the AdipoR1/AMPK pathway interference with the cell metabolic reorientation induced and required by the virus. We're still in the early stages of understanding how a cell's metabolic orientation can exert a pro- or antiviral effect, paving the way for antiviral mechanisms that need to be further deciphered.
Line 334:
Unfortunately, and even if the literature on the discovery of antivirals selected from high-throughput screening of compound libraries, from observation of traditional pharmacopoeia usages, notably based on plant and marine biodiversity[45,46], or from de novo design, is plethoric, it has to be admitted that, to date, few therapeutic strategies have resulted in conclusive clinical phases[47]+ Investigational drugs for the treatment of Zika virus infection: a preclinical and clinical update Han and Mesplède Expert Opin Investig Drugs. 2018 DOI: 10.1080/13543784.2018.1548609)
Line 343:
There are several possible strategies. A first approach is to test molecules with proven antiviral activity, already validated and already used for other viral infections. A potentially pan-viral curative treatment regimen combining drugs targeting several proteins (the viral replicase and proteases) of the hepatitis C virus (HCV), a Flaviviridae member related to ZIKV has been developed with success[48,49]. Another approach would turn to molecules with cytoprotective effects used in CNS pathologies not necessarily of infectious origin. This is why we considered adiponectin, and more specifically its pharmacological analog AdipoRon, as a potential candidate of interest. These molecules, already known to exert neuroprotective activity fit the essential criteria set out above[14,50]. Moreover, while AMPK dysregulation and metabolic dysfunction have been documented in neurodegenerative processes, activation of the AdipoR1/AMPK pathway by AdipoRon has been shown to be cytoprotective and therapeutic in neuroinflammation following intracerebral hemorrhage (Zheng et l. AdipoRon Attenuates Neuroinflammation After Intracerebral Hemorrhage Through AdipoR1-AMPK Pathway. Neuroscience. 2019 doi: 10.1016/j.neuroscience.2019.05.060.; Yu et al. AdipoRon Protects Against Secondary Brain Injury After Intracerebral Hemorrhage via Alleviating Mitochondrial Dysfunction: Possible Involvement of AdipoR1-AMPK-PGC1α Pathway. Neurochem Res. 2019 doi: 10.1007/s11064-019-02794-5.).
Line 458
It should be noted that several studies describe antiviral drugs that target cellular factors, notably AMPK. PF-06409577, an AMPK activator that promotes its phosphorylation and leads to modifications in the host cell lipid metabolism, has been identified as being able to significantly impede the replication of various flaviviruses such as ZIKV (DOI: 10.1128/AAC.00360-18). AdipoRon induces AMPK activation, via the adipoR1 signaling pathway (Figure 6). As with PF-06409577, the specific reduction in levels of various lipids important for flavivirus multiplication, subsequent to this signaling pathway and an activated AMPK, would be at the origin of viral restriction (doi: 10.1016/j.virol.2015.06.015.).
- Emphasize the novelty of the study and its contribution to the field. Clearly articulate how the identification of the AdipoR1/AMPK pathway's antiviral potential against ZIKV adds new insights to existing knowledge. Highlight the significance of targeting host cell responses and metabolic orientation as a promising avenue for antiviral drug development.
This was an important point, and we hope that the various additions made in the discussion of the manuscript have met the Reviewer 1's concerns. We addressed the AdipoR1/AMPK pathway’s antiviral potential in the response to point 9, line 412 (see above) and emphasized the novelty and contribution to the field in the conclusions.
Line 526:
In conclusion, we are still in the early stages of understanding how a cell's metabolic orientation can exert a pro- or antiviral effect. This opens the way to antiviral mechanisms that need to be further deciphered. Our study has the merit of identifying the antiviral potential of the AdipoR1/AMPK/PPAR-α/CPT1 signaling pathway against ZIKV. Targeting host cell responses and their metabolic orientation is therefore a promising strategy for the development of broad-spectrum antiviral drugs that can effectively combat various viruses sharing common strategies in interacting with host cells, and in particular opens up innovative prospects for the treatment of ZIKV specific pathologies.

Reviewer 3 Report
Comments and Suggestions for Authors
Current study by Safadi et al. examines the effect of AdipoRon as an antiviral agent against Zika virus, using A549 epithelial cells as model for ZIKV infection. Authors show that a pre or co-treatment with AdipoRon revealed promising antiviral effects in A549 epithelial cells infected with ZIKV. Purified adiponectin and its pharmacological analog, AdipoRon promote the expression of adiponectin receptors in A549 cells, which mediates the antiviral effects of these compounds against ZIKV. The manuscript is very well written with appropriate attention to previously published literature. I have some minor comments:
The authors have hypothesized that the antiviral effect of adipoRon on ZIKV is mediated by adiponectin signaling. The evaluation of anti-ZIKV activity of adipoRon in ADIPOR1 and ADIPOR2-deficient ZIKV-permissive cell lines would clarify the molecular mechanism of antiviral effect exerted by adipoRon on ZIKV and further strengthen the manuscript.
Co- and Post-infection treatment of ZIKV-infected cells with adipoRon showed reduction in %ZIKV-E positive cells (Figure 3B-C) and ZIKV RNA in the cells (Figure 3 D), but apparently, no significant cell protection was observed as assessed by LDH release assay (Figure 3E). What is the possible explanation for this observation?
Naming the AdipoRon post-infection treatment conditions as adipoRon 24h post ZIKV and adipoRon 40h post ZIKV instead of ZIKV 24h pre adipoRon and ZIKV 40h pre adipoRon would make the manuscript easier to follow.
Section 3.5, figure 5B demonstrates that AdipoRon+ZIKV treatment shows downregulation of both ADIPOR1 and CPT1 gene relative to mock-infection or AdipoRon treatment but AdipoRon is still effective with AdipoRon+ZIKV treatment (observed in Figure 3B and 3C). To attribute the effect of AdipoRon to the upregulation of ADIPOR1 and CPT1, it would be better to show the statistical difference in ADIPOR1 and CPT1 expression levels between ZIKV 48 h and other treatment conditions (Figure 5B).
Typographical errors:
Page 3, Line 107: harvested> cultured
Page 12, line 399: delete “whatever”
Author Response
Current study by Safadi et al. examines the effect of AdipoRon as an antiviral agent against Zika virus, using A549 epithelial cells as model for ZIKV infection. Authors show that a pre or co-treatment with AdipoRon revealed promising antiviral effects in A549 epithelial cells infected with ZIKV. Purified adiponectin and its pharmacological analog, AdipoRon promote the expression of adiponectin receptors in A549 cells, which mediates the antiviral effects of these compounds against ZIKV. The manuscript is very well written with appropriate attention to previously published literature. I have some minor comments:
We would like to thank reviewer 2 for his/her careful and critical reading of our manuscript.
Taking into account your comments, we offer a few clarifications which have led to modifications in the body of the manuscript and in some figures. Our responses to the points raised are indicated in red below each point. The same applies to changes made to the text of the manuscript.
The authors have hypothesized that the antiviral effect of adipoRon on ZIKV is mediated by adiponectin signaling. The evaluation of anti-ZIKV activity of adipoRon in ADIPOR1 and ADIPOR2-deficient ZIKV-permissive cell lines would clarify the molecular mechanism of antiviral effect exerted by adipoRon on ZIKV and further strengthen the manuscript.
We totally agree with the reviewer. Experiments using siRNA shutdown or ADIPOR-deficient cell lines could provide valuable additional information on the antiviral molecular mechanisms of this signaling pathway. The need to move on to in vivo models would also make it possible to decipher these mechanisms, for example, working with a knockout (KO) cell model for adipoR1 (DOI: 10.1172/jci.insight.156301) or other effectors of the AdipoR1/AMPK pathway (doi: 10.1016/j.celrep.2020.108092, doi: 10.1038/s42255-023-00835-6). Inhibiting this pathway using specific inhibitors, could also have added valuable depth to our observations. Unfortunately, we were unable to pursue these approaches for the moment, due to resource constraints. However, we aim to explore these avenues in the near future to further strengthen our findings.
Taking your comment into account, we have added this relevant point to the discussion section.
Line 516
In addition, we need to develop in vivo investigations in mice (Mouse models of Zika virus transplacental transmission. (Li et al., Antiviral Res. 2023 doi: 10.1016/j.antiviral.2022.105500.) a choice model offering knockout (KO) mutations in various genes related to the AdipoR1/AMPK/CPT1a signaling pathway (DOI: 10.1172/jci.insight.156301 ; DOI:10.2337/db06-1432 ; doi: 10.1016/j.celrep.2020.108092 ; doi: 10.1038/s42255-023-00835-6). This step will be a prerequisite for moving on to clinical trials.
Co- and Post-infection treatment of ZIKV-infected cells with adipoRon showed reduction in %ZIKV-E positive cells (Figure 3B-C) and ZIKV RNA in the cells (Figure 3 D), but apparently, no significant cell protection was observed as assessed by LDH release assay (Figure 3E). What is the possible explanation for this observation?
For the post-infection treatment of ZIKV-infected cells with adipoRon, the measure following the 40 hours of infection followed by 8 hours of adipoRon gives a value that is coherent, as the infection was not reduced by this short timing of treatment. We do agree, however, that the non-significant difference in LDH release obtained with the 48-hours co-treatment condition raises questions. Since co-treatment halved the percentage of infected cells, we can only speculate that there may be a threshold of antiviral effect to be reached in order to have a cytoprotective effect. Another possible explanation is based on what we have shown in the figure 5B, as we demonstrated that infection and cotreatment with AdipoRon did not significantly counterbalance the decrease in CPT1 expression induced by ZIKV alone.
Yet, it is known that activation of the AMPK/CPT1a pathway plays a role in inhibiting apoptosis (doi: 10.1038/cddis.2016.388.). The antagonistic effect of ZIKV in the co-treatment condition would limit cytoprotection, which could explain the LDH values despite the reduced percentage of infection.
Naming the AdipoRon post-infection treatment conditions as adipoRon 24h post ZIKV and adipoRon 40h post ZIKV instead of ZIKV 24h pre adipoRon and ZIKV 40h pre adipoRon would make the manuscript easier to follow.
We have modified accordingly, line 167, 168 and in Figures.
Section 3.5, figure 5B demonstrates that AdipoRon+ZIKV treatment shows downregulation of both ADIPOR1 and CPT1 gene relative to mock-infection or AdipoRon treatment but AdipoRon is still effective with AdipoRon+ZIKV treatment (observed in Figure 3B and 3C). To attribute the effect of AdipoRon to the upregulation of ADIPOR1 and CPT1, it would be better to show the statistical difference in ADIPOR1 and CPT1 expression levels between ZIKV 48 h and other treatment conditions (Figure 5B).
Regarding your request, the statistical comparisons have been changed in Figure 5B
Typographical errors:
Page 3, Line 107: harvested> cultured
Page 12, line 399: delete “whatever”
Thank you for your careful attention. These mistakes were corrected.

Round 2
Reviewer 1 Report
Comments and Suggestions for Authors
No further comments
Reviewer 2 Report
Comments and Suggestions for Authors
No comments
Comments on the Quality of English LanguageMinor editing of English language required